# The association of skin autofluorescence with cardiovascular events and all-cause mortality in persons with chronic kidney disease stage 3: A prospective cohort study

Adam Shardlow[1], Natasha J. McIntyre[1], Nitin V. Kolhe[2], Laura B. Nellums[3], Richard J. Fluck[2], Christopher W. McIntyre[4,5], Maarten W. Taal[1,2]*

1 Centre for Kidney Research and Innovation, Division of Medical Sciences and Graduate Entry Medicine, School of Medicine, University of Nottingham, Derby, United Kingdom, 2 Renal Unit, Royal Derby Hospital, Derby, United Kingdom, 3 Division of Epidemiology and Public Health, School of Medicine, University of Nottingham, Nottingham, United Kingdom, 4 Division of Nephrology, Schulich School of Medicine and Dentistry, University of Western Ontario, London, Ontario, Canada, 5 Department of Nephrology, The Victoria Hospital, London Health Sciences Centre, London, Ontario, Canada

* maarten.taal1@nhs.net

## Abstract

### Background

Tissue advanced glycation end product (AGE) accumulation has been proposed as a marker of cumulative metabolic stress that can be assessed noninvasively by measurement of skin autofluorescence (SAF). In persons on haemodialysis, SAF is an independent risk factor for cardiovascular events (CVEs) and all-cause mortality (ACM), but data at earlier stages of chronic kidney disease (CKD) are inconclusive. We investigated SAF as a risk factor for CVEs and ACM in a prospective study of persons with CKD stage 3.

### Methods and findings

Participants with estimated glomerular filtration rate (eGFR) 59 to 30 mL/min/1.73 m$^2$ on two consecutive previous blood tests were recruited from 32 primary care practices across Derbyshire, United Kingdom between 2008 and 2010. SAF was measured in participants with CKD stage 3 at baseline, 1, and 5 years using an AGE reader (DiagnOptics). Data on hospital admissions with CVEs (based on international classification of diseases [ICD]-10 coding) and deaths were obtained from NHS Digital. Cox proportional hazards models were used to investigate baseline variables associated with CVEs and ACM. A total of 1,707 of 1,741 participants with SAF readings at baseline were included in this analysis: The mean (± SD) age was 72.9 ± 9.0 years; 1,036 (60.7%) were female, 1,681 (98.5%) were of white ethnicity, and mean (±SD) eGFR was 53.5 ± 11.9 mL/min/1.73 m$^2$. We observed 319 deaths and 590 CVEs during a mean of 6.0 ± 1.5 and 5.1 ± 2.2 years of observation, respectively. Higher baseline SAF was an independent risk factor for CVEs (hazard ratio [HR] 1.12 per SD, 95% CI 1.03–1.22, $p = 0.01$) and ACM (HR 1.16, 95% CI 1.03–1.30, $p = 0.01$). Additionally, increase in SAF over 1 year was independently associated with subsequent CVEs (HR 1.11

**Data Availability Statement:** We are unable to make the data available in a public repository, within the manuscript itself, or uploaded as supplementary information because: 1. This is not permitted by our organisation's research governance policy. 2. It would be in breach of UK Data Protection legislation. 3. It is specifically not permitted in our Data Sharing Agreement with NHS Digital. Anonymised data can be made available only to researchers who meet the conditions of the ethics approval and research governance policy that applies to this study. Researchers may apply for data access by contacting Dr Teresa Grieve, Research and Development Deputy Director, University Hospitals of Derby and Burton NHS Foundation Trust (teresa.grieve@nhs.net).

**Funding:** The RRID study was funded by a Research Project Grant (R302/0713) from the Dunhill Medical Trust (https://dunhillmedical.org.uk) awarded to MWT. Previous study funding includes a joint British Renal Society (https://britishrenal.org) and Kidney Research UK (https://kidneyresearchuk.org) fellowship, awarded to NJM, and an unrestricted educational grant from Roche Products Ltd (https://www.roche.co.uk) awarded to MWT. The funders had no role in study design, data collection and analysis, decision to publish, or preparation of the manuscript.

**Competing interests:** I have read the journal's policy and the authors of this manuscript have the following competing interests: MWT is an Academic Editor for PLOS Medicine.

**Abbreviations:** ACM, all-cause mortality; AGE, advanced glycation end product; AU, arbitrary units; BMI, body mass index; CKD, chronic kidney disease; CKD-EPI, chronic kidney disease epidemiology; CRP, C reactive protein; CVD, cardiovascular disease; CVE, cardiovascular event; DBP, diastolic blood pressure; eGFR, estimated glomerular filtration rate; HD, haemodialysis; HR, hazard ratio; hsCRP, high-sensitivity C reactive protein; ICD, international classification of diseases; IDMS, isotope dilution mass spectrometry; KDIGO, kidney disease improving global outcomes; MDRD, modification of diet in renal disease; NIHR, National Institute for Health Research; OR, odds ratio; PD, peritoneal dialysis; RAGE, receptor for AGE; RRID, Renal Risk in Derby; SAF, skin autofluorescence; SBP, systolic blood pressure; TIA, transient ischaemic attack; UACR, urine albumin to creatinine ratio.

per SD, 95% CI 1.00–1.22; $p = 0.04$) and ACM (HR 1.24, 95% CI 1.09–1.41, $p = 0.001$). We relied on ICD-10 codes to identify hospital admissions with CVEs, and there may therefore have been some misclassification.

## Conclusions

We have identified SAF as an independent risk factor for CVE and ACM in persons with early CKD. These findings suggest that interventions to reduce AGE accumulation, such as dietary AGE restriction, may reduce cardiovascular risk in CKD, but this requires testing in prospective randomised trials. Our findings may not be applicable to more ethnically diverse or younger populations.

## Author summary

### Why was this study done?

- Advanced glycation end products (AGEs) are chemical compounds that play a role in health problems associated with aging, diabetes, and heart disease.

- The kidneys play a role in removing AGEs; therefore, people with kidney disease can develop accumulation of AGEs over time.

- The measurement of skin autofluorescence (SAF) is a noninvasive method to assess AGE accumulation.

- An important previous study found that people requiring dialysis have high SAF levels and that these are strong predictors of a higher risk of death from heart disease or any cause, but SAF has not been as well studied in people with milder forms of kidney disease who are also at higher risk of heart disease.

### What did the researchers do and find?

- A total of 1,707 people with relatively mild chronic kidney disease (CKD, stage 3) and average age of 73 years were enrolled into this study from 32 primary care practices across Derbyshire, United Kingdom between 2008 and 2010.

- During an observation period of 5 to 6 years, we found that a higher SAF level at enrolment was associated with a 12% higher risk of having a heart attack, heart failure, or stroke and a 16% higher risk of death from any cause.

- Additionally, an increase in SAF level over 1 year was associated with a 11% higher risk of having a heart attack, heart failure, or stroke and a 24% higher risk of death from any cause.

### What do these findings mean?

- Higher SAF levels are an independent risk factor for heart attacks, heart failure, stroke, or death from any cause in people with mild chronic kidney disease, though the risk seems lower than in people requiring dialysis.

- We should now explore ways to lower AGE levels in people with kidney disease, which may include adaptations to reduce the amount of AGEs in the diet.

- Because our study population was predominantly elderly and of white ethnicity, our findings may not be directly applicable to more ethnically diverse or younger populations.

## Introduction

Advanced glycation end products (AGEs) are cross-linking compounds that play a role in the pathogenesis of aging, diabetic microvascular complications, and cardiovascular disease (CVD). Glycation and oxidation of amino groups on proteins results in AGE formation through a series of nonenzymatic reactions termed the Maillard reaction. AGEs may also be generated more rapidly by reactions with α-dicarbonyls that are produced during oxidative stress [1,2]. Tissue accumulation of AGEs has therefore been proposed as a marker of cumulative 'metabolic stress'. Exogenous AGEs from food (particularly food cooked at high temperatures) [3] and smoking [4] as well as decreased renal excretion in chronic kidney disease (CKD) [5] may also contribute to AGE accumulation. Skin autofluorescence (SAF) measurement has been developed as a noninvasive marker of AGE accumulation in the skin and has been validated using skin biopsy samples [6]. Measurement of SAF can be carried out quickly and easily using portable equipment and may therefore be useful as a noninvasive measure to risk-stratify persons with CKD.

CKD is associated with a marked increase in cardiovascular events (CVEs), but risk assessment tools developed in general population studies tend to underestimate this risk [7], in part because of the importance of nontraditional risk factors. AGE accumulation may be one such risk factor that was suggested by a landmark paper reporting that higher SAF was a strong and independent risk factor for cardiovascular and all-cause mortality (ACM) in persons on haemodialysis [6]. We have previously reported that in persons with earlier stage CKD, higher SAF was associated with multiple risk factors for CVD in a cross-sectional analysis [8] and was a risk factor for increased ACM in univariable but not fully adjusted multivariable models [9]. After a longer observation period in the same cohort, we sought to investigate whether SAF is an independent risk factor for CVEs and ACM persons with CKD stage 3, cared for in primary care.

## Methods

The Renal Risk in Derby (RRID) study is a prospective cohort study of persons with CKD stage 3 recruited from primary care across Derbyshire. A detailed description of methods has been published previously [8,10]. The study was conducted according to a prospective protocol and is reported in keeping with the Strengthening the Reporting of Observational Studies in Epidemiology (STROBE) guideline (S1 STROBE Checklist). Please see S1 Protocol for the latest version of the study protocol.

### Participants

Participants were individually recruited from 32 primary care practices across Derbyshire, United Kingdom between 2008 and 2010. Participating practices were asked to invite persons over 18 years of age with CKD stage 3 from CKD registers. Eligible persons were selected using estimated glomerular filtration rate (eGFR) values calculated using the 4-variable modification of diet in renal disease (MDRD) equation modified for use with isotope dilution mass spectrometry (IDMS)-standardised creatinine measurement. Two eGFR readings more than

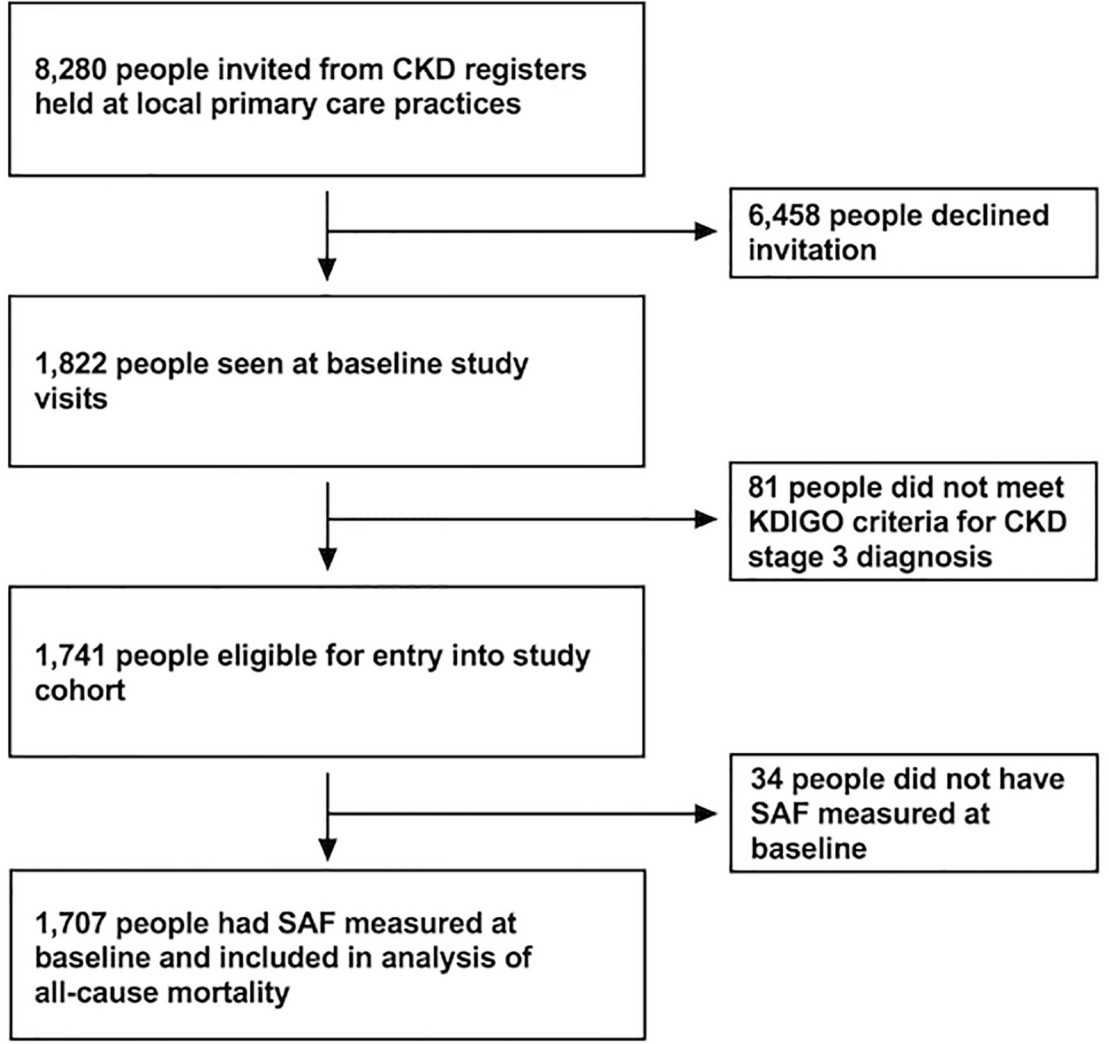

**Fig 1. Flow chart showing the numbers of participants involved at each stage of the study.** CKD. chronic kidney disease; KDIGO, kidney disease improving global outcomes; SAF, skin autofluorescence.

90 days apart in the range 30 to 59 mL/min/1.73 m$^2$ were required to be eligible. Those with a previous renal or other solid organ transplant, with an expected life-expectancy of less than 1 year, or who were unable to attend the baseline visit in person were excluded. A total of 8,280 persons were invited to take part in the study by post; 1,822 persons attended for baseline visits, of whom 1,741 were suitable for recruitment. Baseline SAF measurements were obtained in 1,707 participants, and these are included in the analysis (Fig 1). Participants gave written informed consent prior to the baseline assessment. The study was approved by Nottingham Research Ethics Committee 1 (reference number 08/H0403/16) and is included in the National Institute for Health Research (NIHR) clinical research portfolio (NIHR Study ID 6632). The study follows the principles of Good Clinical Practice and the Declaration of Helsinki.

### Data collection

Study visits took place at the participants' primary care practice at baseline, 1, and 5 years. Prior to each visit, participants completed a background questionnaire covering demographics,

medical history, smoking history, and medication history (see S1 Questionnaire for the data questionnaire). Responses were reviewed during the study visit and clarified as needed. Past medical history of CVD was defined as participant-reported previous myocardial infarction, stroke, transient ischaemic attack (TIA), amputation or revascularisation for peripheral vascular disease, and abdominal aortic aneurysm. Participants provided three consecutive early morning urine samples, stored in a refrigerator prior to the study visit. Urine samples were analysed for albumin to creatinine ratio. Blood samples for biochemistry and haematology were taken from each participant. Participants were asked to avoid eating meat for 12 hours prior to their study visit to avoid confounding their serum creatinine results.

At each study visit, height and weight were measured. Three blood pressure measurements that differed by less than 10% were obtained after at least 5 minutes rest using an automatic oscillometric device (UA-767 Plus 30, A&D Medical). The average of three readings was used for analysis.

### Laboratory methods

Blood and urine samples were analysed at a single clinical laboratory at the Royal Derby Hospital. Creatinine was measured using the Jaffe reaction and was standardised to IDMS methods. eGFR was calculated using the MDRD equation [11] at the time of recruitment, but for analysis, this was changed to the more accurate chronic kidney disease epidemiology (CKD-EPI) equation [12], published after recruitment commenced. Additionally, serum was analysed for standard electrolytes and bone mineral profile. Urinary albumin was measured using an immunoturbidimetric assay ('Tina-quant', Roche Diagnostics, Mannheim, Germany) on a Roche Modular system. Urine albumin to creatinine ratio (UACR) was measured on three urine samples from each participant, and a mean value was used for analysis. Serum high-sensitivity C reactive protein (CRP) (hsCRP™, Roche Diagnostics, Newhaven, UK) was measured using a Roche Modular P Analyser (Roche Diagnostics) at The Binding Site Group laboratories, Birmingham, UK.

### SAF

SAF was measured using an AGE reader (DiagnOptics Technologies BV, Groningen, The Netherlands). The AGE reader provides a noninvasive measure of skin AGE levels that has been validated using data from skin biopsies. A light source emitting light at a wavelength of 320 to 400 nm excites fluorescent moieties in compounds in the skin to produce fluorescence at wavelength 420 to 600 nm (peak 440 nm). The output represents the ratio between autofluorescence in the range 420 to 600 nm and excitation light in the range 320 to 400 nm and is reported in arbitrary units (AU). The AGE reader is not able to obtain valid SAF readings when the skin reflectivity is lower than 6%. Persons with dark skin colour (Fitzpatrick skin colour type V–VI) were therefore excluded from this aspect of the study ($n$ = 17). Technical failure prevented SAF readings in a further 17 participants. We have previously reported that SAF readings have good reproducibility and repeatability (coefficient of variation of 7%–8%) [8]. Three SAF measurements were taken from the ventral (anterior) surface of the forearm of each participant, avoiding any tattoos or heavily pigmented areas of skin, and the average was used for analysis.

### Outcomes

The outcomes of interest for this analysis were fatal and nonfatal CVEs and ACM. Data on all deaths and hospital admissions from date of recruitment to 31 December 2015 were obtained from NHS Digital under a data sharing agreement. NHS Digital holds data on all deaths (from

death certificates) and coding data on all hospital admissions in England and Wales. Three investigators (AS, RJF, and MWT) independently classified cause of death as cardiovascular or noncardiovascular. Differences were resolved by discussion. CVEs were defined as any cardiovascular death or hospitalisation that included myocardial infarction, stroke, TIA, cardiac failure, revascularisation, or peripheral vascular disease identified from ICD-10 codes in any of the diagnoses.

## Statistical methods

Data are presented as mean ± SD or median (interquartile range) depending on distribution. Normally distributed continuous variables were compared across tertiles using ANOVA. Nonparametrically distributed variables were compared using the Kruskal–Wallis test. Categorical variables were compared using chi squared tests. Missing data were omitted from analyses.

We constructed multilevel mixed-effects models using the mixed command in Stata 15 to investigate factors associated with SAF as a repeated measure at baseline, Year 1, and Year 5. Cox proportional hazards models were constructed to investigate variables associated with time to death from any cause or time to hospitalisation with a CVE or cardiovascular death. All variables that evidenced a significant univariable association ($p < 0.05$) with the outcome of interest were subsequently entered into multivariable models. Model 1 included demographic and past medical history variables; Model 2 added blood pressure, body mass index (BMI), eGFR, and UACR; Model 3 added all remaining laboratory variables including high-sensitivity C reactive protein (hsCRP). Hazard ratios for continuous variables are expressed per SD change. To facilitate this, continuous variables that were not normally distributed (UACR and hsCRP) were logarithmically transformed prior to inclusion in Cox proportional hazards models.

Analyses were conducted using SPSS version 24 (IBM corporation, NY, USA) and Stata 15 (StataCorp LLC, Texas, USA). $p < 0.05$ was regarded as statistically significant.

## Results

### Baseline characteristics

Baseline SAF readings were obtained in 1,707 persons and are included in this analysis (98% of a total of 1,741 in the RRID cohort). The mean age of those included was 72.9 ± 9.0 years, 1,036 (60.7%) were female, 1,681 (98.5%) were of white ethnicity, mean eGFR was 53.5 ± 11.9 mL/min/1.73 m$^2$. Baseline characteristics (including the number of participants with complete data) are presented in Table 1 by tertile of SAF. Participants in the highest tertile of SAF were more likely to be male, either a current or previous smoker, have type 1 or 2 diabetes, and have a past history of CVD. Additionally, higher age, systolic blood pressure (SBP), UACR, serum uric acid, and hsCRP were associated with a higher tertile of baseline SAF. Lower diastolic blood pressure (DBP), eGFR, haemoglobin, serum albumin, and total cholesterol were associated with higher tertile of baseline SAF.

### Change in SAF over time

Among 948 participants who had SAF measured at baseline and Year 5, no change in mean SAF was observed over time (baseline: 2.6 ± 0.6 AU; Year 1: 2.5 ± 0.5 AU, Year 5: 2.7 ± 0.6 AU; $p$ = 0.1). Multilevel mixed-effects models showed that greater age, male sex, diabetes, previous CVD, current or previous smoking, lower eGFR, lower serum albumin, and lower haemoglobin were independently associated with higher SAF in repeated measures over the follow-up period (Table 2).

**Table 1. Baseline characteristics by tertile of SAF.**

| Variable | Number[a] | Lowest Tertile (*n* = 560) | Middle Tertile (*n* = 575) | Highest Tertile (*n* = 572) | *P* value |
|---|---|---|---|---|---|
| **Baseline SAF (AU)** | 1,707 | 2.1 ± 0.2 | 2.7 ± 0.1 | 3.4 ± 0.4 | <0.001 |
| **Baseline age (years)** | 1,707 | 70.7 ± 9.9 | 73.0 ± 8.6 | 74.9 ± 8.1 | <0.001 |
| **Female sex** | 1,707 | 360 (64.3) | 359 (62.4) | 317 (55.4) | 0.005 |
| **Diabetes** | 1,707 | 50 (8.9) | 82 (14.3) | 152 (26.6) | <0.001 |
| **Previous CVD** | 1,707 | 88 (15.7) | 127 (22.1) | 164 (28.7) | <0.001 |
| **Smoking status** | | | | | |
| Current | 1,707 | 18 (3.2) | 18 (3.1) | 43 (7.5) | <0.001 |
| Previous | 1,707 | 242 (43.2) | 283 (49.2) | 327 (57.2) | <0.001 |
| **BMI (kg/m$^2$)** | 1,706 | 28.7 ± 4.9 | 29.3 ± 5.3 | 29.1 ± 5.2 | 0.2 |
| **Systolic BP (mmHg)** | 1,707 | 133 ± 18 | 133 ± 18 | 136 ± 19 | 0.006 |
| **Diastolic BP (mmHg)** | 1,707 | 74 ± 11 | 73 ± 11 | 71 ± 11 | <0.001 |
| **eGFR (mL/min/1.73 m$^2$)** | 1,707 | 57.1 ± 11.2 | 53.6 ± 11.0 | 49.9 ± 12.2 | <0.001 |
| **UACR (mg/mmol)** | 1,704 | 0.2 (0.0–0.9) | 0.3 (0.0–1.3) | 0.6 (0.0–3.0) | <0.001 |
| **Albumin (g/l)** | 1,704 | 41.1 ± 2.9 | 41.0 ± 3.4 | 40.2 ± 3.1 | <0.001 |
| **Phosphate (mmol/l)** | 1,676 | 1.11 ± 0.17 | 1.11 ± 0.18 | 1.11 ± 0.18 | 0.90 |
| **Calcium (mmol/l)** | 1,696 | 2.38 ± 0.10 | 2.37 ± 0.10 | 2.38 ± 0.10 | 0.4 |
| **Bicarbonate (mmol/l)** | 1,688 | 25.6 ± 2.5 | 25.6 ± 2.7 | 25.3 ± 2.9 | 0.06 |
| **Urate (umol/l)** | 1,697 | 374 ± 88 | 390 ± 96 | 387 ± 88 | 0.005 |
| **Total Cholesterol (mmol/l)** | 1,698 | 4.9 ± 1.2 | 4.8 ± 1.2 | 4.6 ± 1.1 | <0.001 |
| **HDL Cholesterol (mmol/L)** | 1,698 | 1.47 ± 0.43 | 1.47 ± 0.45 | 1.42 ± 0.42 | 0.08 |
| **Haemoglobin (g/dl)** | 1,702 | 13.5 ± 1.3 | 13.3 ± 1.4 | 12.9 ± 1.5 | <0.001 |
| **hsCRP (mg/L)** | 1,706 | 2.06 (1.04–4.02) | 2.10 (1.09–4.50) | 2.56 (1.30–5.26) | 0.001 |

Data are presented as mean ± SD, number (percentage), or median (interquartile range).

*P* values for trend across tertiles by ANOVA, Chi squared test, or Kruskal–Wallis test.

[a]Number of complete data for each variable.

**Abbreviations**: AU, arbitrary units; BMI, body mass index; BP, blood pressure; CVD, cardiovascular disease; eGFR, estimated glomerular filtration rate; HDL, high density lipoprotein; hsCRP, high-sensitivity C reactive protein; SAF, skin autofluorescence; UACR, urine albumin to creatinine ratio

## CVEs

We observed 590 CVEs during 5.1 ± 2.2 years of observation, of which 105 were fatal. Kaplan–Meier analysis showed a progressive increase in CVEs across tertiles of baseline SAF (Fig 2). Additionally, multivariable Cox proportional hazards analysis identified SAF at baseline as an independent risk factor for time to first CVE (HR 1.12 per SD increase, 95% CI 1.03–1.22, *p* = 0.01) together with age, male sex, history of previous CVD, higher UACR, lower DBP, lower serum albumin, and higher hsCRP (Table 3). In subgroup analyses, baseline SAF remained independently associated with nonfatal CVEs (S1 Table) but an association with fatal CVEs in the univariable analysis and initial multivariable analysis (Model 1) was not maintained after full multivariable analysis (S2 Table). In a further subgroup analysis of participants with no events during the first year, change in SAF over 1 year was independently associated with CVEs (HR 1.11 per SD increase, 95% CI 1.00–1.22; *p* = 0.04) though the association with baseline SAF was attenuated (HR 1.12 per SD increase, 95% CI 1.00–1.27; *p* = 0.06; S3 Table).

## ACM

We observed 319 deaths (ACM) during 6.0 ± 1.5 years of observation. Kaplan–Meier analysis showed a progressive increase in risk of ACM across tertiles of baseline SAF (Fig 3).

**Table 2. Multilevel mixed-effects models for associations with SAF as a repeated measure at baseline, Year 1, and Year 5.**

| Variable | Univariable analysis | | Multivariable analysis (*n* = 1,668) | |
|---|---|---|---|---|
| | Coefficient (95% CI) | *p*-value | Coefficient (95% CI) | *p*-value |
| Age | 0.01 (0.01–0.02) | <0.001 | 0.01 (0.00–0.01) | <0.001 |
| Male sex | 0.16 (0.10–0.21) | <0.001 | 0.11 (0.05 to −0.16) | <0.001 |
| Diabetes | 0.34 (0.28–0.41) | <0.001 | 0.23 (0.16–0.30) | <0.001 |
| Previous CVD | 0.22 (0.16–0.29) | <0.001 | 0.11 (0.05–0.17) | <0.001 |
| Previous smoker | 0.15 (0.09–0.20) | <0.001 | 0.10 (0.05–0.15) | <0.001 |
| Current smoker | 0.28 (0.16–0.40) | <0.001 | 0.41 (0.29–0.52) | <0.001 |
| eGFR | −001 (−0.02 to −0.01) | <0.001 | −0.01 (−0.01 to −0.00) | <0.001 |
| UACR | 0.00 (0.00–0.00) | 0.02 | −0.00 (−0.00 to 0.00) | 0.84 |
| Total cholesterol | −0.09 (−0.11 to −0.07) | <0.001 | −0.02 (−0.04 to −0.00) | 0.08 |
| Albumin | −0.02 (−0.03 to −0.02) | <0.001 | −0.01 (−0.02 to −0.00) | 0.02 |
| Bicarbonate | −0.02 (−0.03 to −0.01) | <0.001 | −0.01 (−0.02 to 0.00) | 0.18 |
| Haemoglobin | −0.08 (−0.10 to −0.07) | <0.001 | −0.06 (−0.08 to −0.04) | <0.001 |
| hsCRP | 0.00 (0.00–0.01) | <0.001 | 0.00 (−0.00 to 0.00) | 0.09 |

**Abbreviations**: CVD, cardiovascular disease, eGFR, estimated glomerular filtration rate; hsCRP, high-sensitivity C reactive protein; SAF, skin autofluorescence; UACR, urine albumin to creatinine ratio

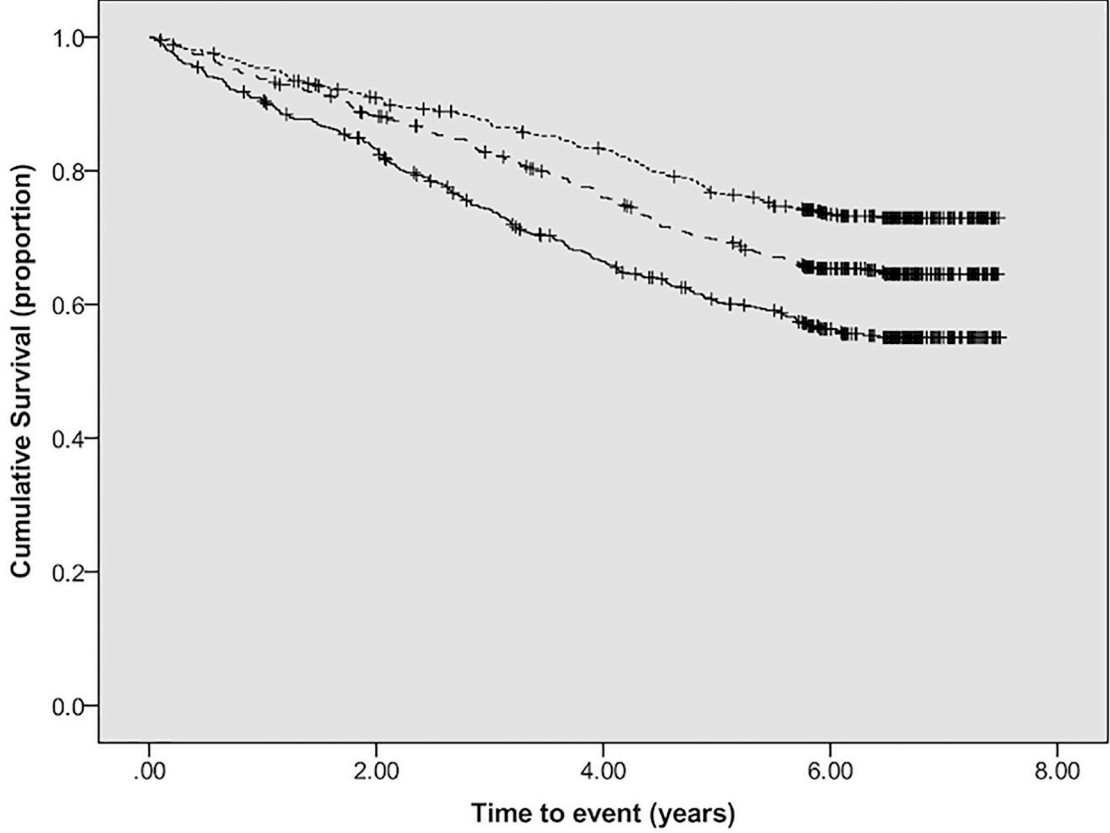

**Fig 2. Kaplan–Meier plot showing CVE free survival by tertiles of SAF (dotted line, lowest tertile; dashed line, middle tertile; solid line, highest tertile; log-rank test: chi-square 41.4; *p* < 0.001).** CVE, cardiovascular event; SAF, skin autofluorescence.

**Table 3. Cox proportional hazards model showing variables associated with time to first CVE.**

| Variable | Univariable | | Model 1 (n = 1,707) | | Model 2 (n = 1,703) | | Model 3 (n = 1,675) | |
|---|---|---|---|---|---|---|---|---|
| | HR (95% CI) | p-value | HR (95% CI) | p-value | HR (95% CI) | p-value | HR (95% CI) | p-value |
| SAF | 1.37 (1.27–1.48) | <0.001 | 1.20 (1.10–1.30) | <0.001 | 1.15 (1.06–1.25) | 0.001 | 1.12 (1.03–1.22) | 0.01 |
| Age | 1.54 (1.41–1.69) | <0.001 | 1.39 (1.26–1.53) | <0.001 | 1.31 (1.18–1.46) | <0.001 | 1.31 (1.17–1.45) | <0.001 |
| Male sex | 1.79 (1.52–2.11) | <0.001 | 1.47 (1.24–1.74) | <0.001 | 1.45 (1.22–1.73) | <0.001 | 1.51 (1.23–1.86) | <0.001 |
| Diabetes | 1.33 (1.08–1.62) | 0.006 | 1.09 (0.89–1.35) | 0.4 | 0.94 (0.76–1.17) | 0.6 | 0.95 (0.76–1.20) | 0.6 |
| Previous CVD | 2.51 (2.12–2.97) | <0.001 | 1.94 (1.63–2.31) | <0.001 | 1.90 (1.59–2.27) | <0.001 | 1.94 (1.62–2.33) | <0.001 |
| Hypertension | 1.76 (1.31–2.36) | <0.001 | 1.28 (0.95–1.73) | 0.1 | 1.14 (0.84–1.56) | 0.4 | 1.23 (0.90–1.69) | 0.2 |
| Ever smoked | 1.44 (1.22–1.69) | <0.001 | 1.14 (0.96–1.35) | 0.2 | 1.10 (0.92–1.31) | 0.3 | 1.08 (0.91–1.29) | 0.4 |
| Systolic BP | 1.06 (0.98–1.15) | 0.2 | | | 1.01 (0.91–1.11) | 0.9 | 1.01 (0.92–1.12) | 0.8 |
| Diastolic BP | 0.79 (0.73–0.86) | <0.001 | | | 0.89 (0.80–0.98) | 0.02 | 0.89 (0.80–0.99) | 0.03 |
| BMI | 1.03 (0.95–1.12) | 0.5 | | | 1.12 (1.02–1.22) | 0.01 | 1.08 (0.99–1.19) | 0.09 |
| eGFR | 0.69 (0.63–0.75) | <0.001 | | | 0.88 (0.80–0.97) | 0.009 | 0.93 (0.84–1.04) | 0.2 |
| UACR (log) | 1.31 (1.20–1.43) | <0.001 | | | 1.15 (1.05–1.26) | 0.002 | 1.12 (1.02–1.22) | 0.02 |
| Albumin | 0.80 (0.74–0.86) | <0.001 | | | | | 0.89 (0.81–0.96) | 0.005 |
| Uric acid | 1.23 (1.14–1.34) | <0.001 | | | | | 1.02 (0.92–1.12) | 0.7 |
| Total cholesterol | 0.80 (0.73–0.87) | <0.001 | | | | | 1.05 (0.95–1.15) | 0.3 |
| HDL cholesterol | 0.79 (0.72–0.87) | <0.001 | | | | | 0.91 (0.82–1.01) | 0.07 |
| Haemoglobin | 0.84 (0.77–0.91) | <0.001 | | | | | 0.94 (0.86–1.03) | 0.2 |
| hsCRP (log) | 1.25 (1.15–1.35) | <0.001 | - | | | | 1.11 (1.02–1.21) | 0.02 |

Hazard ratios for continuous variables are expressed per SD change.

**Abbreviations**: BMI, body mass index; BP, blood pressure; CVD, cardiovascular disease; CVE, cardiovascular event; eGFR, estimated glomerular filtration rate; HDL, high density lipoprotein; HR, hazard ratio; hsCRP, high-sensitivity C reactive protein; SAF, skin autofluorescence; UACR, urine albumin to creatinine ratio

Additionally, multivariable analysis identified SAF at baseline as an independent risk factor for ACM (HR 1.16 per SD increase, 95% CI 1.03–1.30, $p$ = 0.01) together with age, male sex, history of previous CVD, lower eGFR, and higher hsCRP (Table 4). In subgroup analyses, baseline SAF remained independently associated with noncardiovascular deaths (S4 Table), but an association with cardiovascular deaths in the univariable analysis and initial multivariable analysis (Model 1) was not maintained after full multivariable analysis (S2 Table). In a further subgroup analysis of participants who survived beyond the first year, change in SAF over 1 year was independently associated with ACM (HR 1.24 per SD increase, 95% CI 1.09–1.41, $p$ = 0.001) in addition to baseline SAF (HR 1.25 per SD increase, 95% CI 1.07–1.45; $p$ = 0.005; S5 Table).

## Sensitivity analysis

Sensitivity analyses were undertaken after excluding participants with diabetes to test whether associations with SAF could be attributable to higher SAF values in persons with diabetes. Among 1,423 participants without diabetes, higher SAF remained independently associated with time to first CVE (HR 1.13 per SD increase; 95% CI 1.02–1.25; $p$ = 0.02; S6 Table), but the association with ACM was no longer statistically significant (HR 1.08 per SD increase, 95% CI 0.95–1.24; $p$ = 0.3) in fully adjusted models (S7 Table).

## Discussion

We have identified higher SAF as an independent risk factor for CVEs and ACM in a cohort of persons with predominantly early CKD stage 3. Additionally, an increase in SAF over 1 year

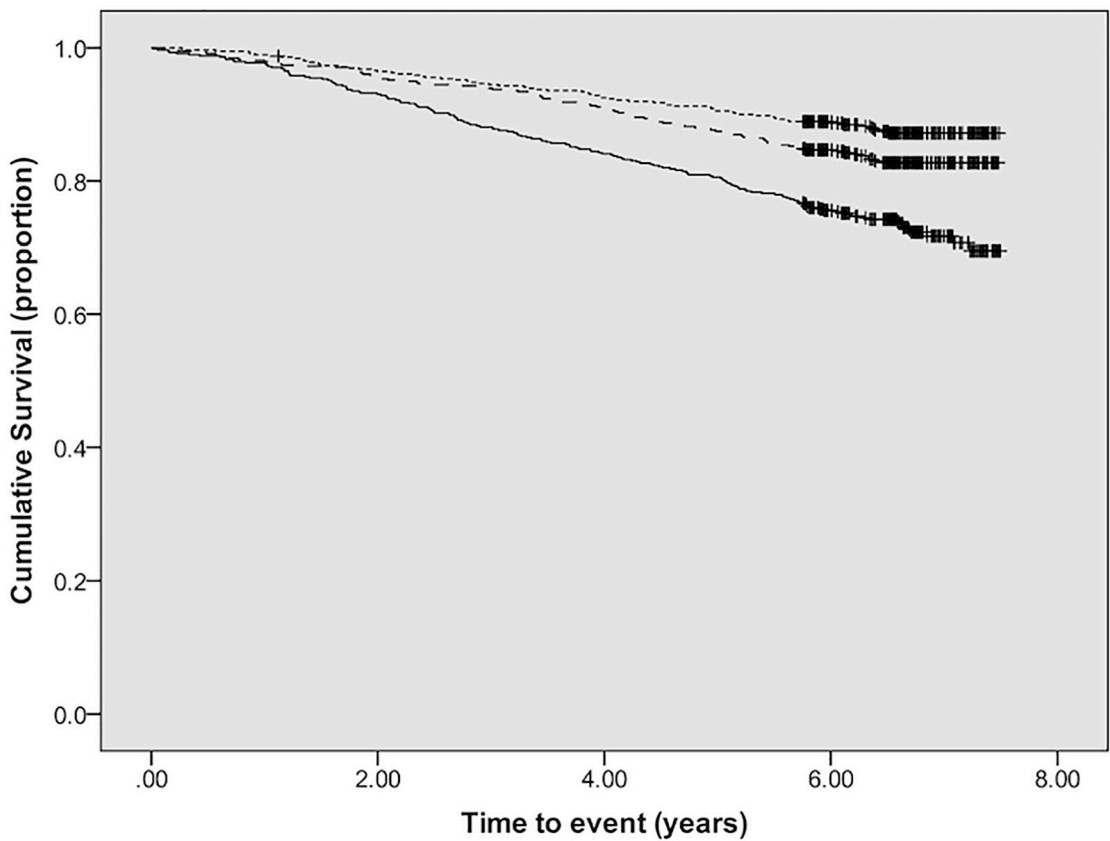

**Fig 3. Kaplan–Meier plot showing survival by tertiles of SAF (dotted line, lowest tertile; dashed line, middle tertile; solid line, highest tertile; log-rank test: chi-square 42.5; $p < 0.001$).** SAF, skin autofluorescence.

was an independent predictor of CVEs and ACM. Our observations extend the findings of previous studies that identified higher SAF as an independent risk factor for cardiovascular mortality and ACM in persons receiving haemodialysis [6] by showing that this association is also present at a much earlier stage of CKD. Sensitivity analyses confirmed that the association with CVEs persisted when persons with diabetes were excluded, though the association with ACM was no longer significant.

Several individual studies and a meta-analysis have confirmed that higher SAF is a strong and independent predictor of cardiovascular mortality and ACM in persons receiving haemodialysis (HD). The first comprehensive study to report this association found that each 1 AU increase in baseline SAF was independently associated with an odds ratio (OR) of 3.9 (95% CI 1.9–8.1) for ACM and an OR of 6.8 (95% CI 2.6–17.5) for cardiovascular mortality in a cohort of 109 persons on HD after 3 years of follow-up [6]. A meta-analysis that included 10 studies of persons with diabetes ($n$ = 2), peripheral arterial disease ($n$ = 1), and CKD ($n$ = 7), reported that higher SAF was associated with an increased risk of cardiovascular mortality and ACM. In a subgroup analysis that included only studies of HD patients, higher SAF was similarly associated with higher risk of cardiovascular mortality (HR 1.97; 95% CI 1.11–3.49) and ACM (HR 2.37; 95% CI 1.72–3.26) [13]. Additionally, in one study, an increase in SAF over 1 year was independently associated with higher subsequent mortality on HD [14]. Similar observations have been reported in persons performing peritoneal dialysis (PD), though a relatively small number of participants precluded multivariable analysis [15,16]. In a mixed study

**Table 4. Cox proportional hazards model showing variables associated with time to death from any cause.**

| Variable | Univariable | | Model 1 (n = 1,707) | | Model 2 (n = 1,703) | | Model 3 (n = 1,675) | |
|---|---|---|---|---|---|---|---|---|
| | HR (95% CI) | p-value | HR (95% CI) | p-value | HR (95% CI) | p-value | HR (95% CI) | p-value |
| SAF | 1.51 (1.37–1.67) | <0.001 | 1.26 (1.13–1.40) | <0.001 | 1.19 (1.06–1.33) | 0.003 | 1.16 (1.03–1.30) | 0.01 |
| Age | 2.46 (2.14–2.83) | <0.001 | 2.26 (1.96–2.61) | <0.001 | 1.93 (1.65–2.25) | <0.001 | 1.90 (1.62–2.23) | <0.001 |
| Male sex | 2.02 (1.62–2.51) | <0.001 | 1.51 (1.20–1.90) | 0.001 | 1.35 (1.07–1.72) | 0.01 | 1.37 (1.03–1.81) | 0.03 |
| Diabetes | 1.45 (1.11–1.89) | 0.007 | 1.18 (0.90–1.55) | 0.2 | 1.09 (0.82–1.45) | 0.6 | 1.08 (0.80–1.46) | 0.6 |
| Previous CVD | 2.38 (1.90–2.98) | <0.001 | 1.66 (1.32–2.10) | <0.001 | 1.64 (1.30–2.08) | <0.001 | 1.62 (1.27–2.06) | <0.001 |
| Hypertension | 1.66 (1.11–2.47) | 0.014 | 1.00 (0.67–1.51) | 1.0 | 0.87 (0.57–1.33) | 0.5 | 0.94 (0.61–1.46) | 0.8 |
| Ever smoked | 1.71 (1.36–2.16) | <0.001 | 1.33 (1.05–1.70) | 0.02 | 1.27 (0.99–1.62) | 0.06 | 1.21 (0.94–1.54) | 0.1 |
| Systolic BP | 1.12 (1.00–1.25) | 0.05 | | | 1.01 (0.89–1.15) | 0.9 | 1.03 (0.90–1.18) | 0.7 |
| Diastolic BP | 0.75 (0.67–0.84) | <0.001 | | | 0.93 (0.81–1.07) | 0.3 | 0.93 (0.80–1.07) | 0.3 |
| BMI | 0.85 (0.75–0.95) | 0.005 | | | 0.94 (0.83–1.07) | 0.3 | 0.91 (0.79–1.04) | 0.2 |
| eGFR | 0.52 (0.46–0.58) | <0.001 | | | 0.73 (0.64–0.84) | <0.001 | 0.77 (0.66–0.89) | 0.001 |
| UACR (log) | 1.47 (1.30–1.66) | <0.001 | | | 1.19 (1.05–1.35) | 0.007 | 1.12 (0.98–1.28) | 0.09 |
| Albumin | 0.78 (0.71–0.85) | <0.001 | | | | | 0.92 (0.81–1.04) | 0.2 |
| Uric acid | 1.30 (1.17–1.45) | <0.001 | | | | | 1.02 (0.90–1.16) | 0.7 |
| Total cholesterol | 0.71 (0.63–0.80) | <0.001 | | | | | 0.94 (0.82–1.09) | 0.4 |
| HDL cholesterol | 0.81 (0.71–0.91) | 0.001 | | | | | 0.98 (0.86–1.13) | 0.8 |
| Haemoglobin | 0.76 (0.68–0.85) | <0.001 | | | | | 0.98 (0.87–1.11) | 0.8 |
| hsCRP (log) | 1.41 (1.27–1.56) | <0.001 | | | | | 1.25 (1.12–1.40) | <0.001 |

Hazard ratios for continuous variables are expressed per SD change.

**Abbreviations**: BMI, body mass index; BP, blood pressure; CVD, cardiovascular disease; eGFR, estimated glomerular filtration rate; HDL, high density lipoprotein; HR, hazard ratio; hsCRP, high-sensitivity C reactive protein; SAF, skin autofluorescence; UACR, urine albumin to creatinine ratio

population of persons with predialysis CKD stage 5 or receiving HD and PD, higher SAF predicted ACM in a multivariable analysis that included traditional Framingham risk factors but was no longer significant after the addition of previous CVD, C reactive protein (CRP), and serum albumin [17]. In persons with earlier stages of CKD, higher SAF has been associated with several aspects of CVD including coronary artery calcification [18], subclinical atherosclerosis [19], and arterial stiffness [8]. Similarly, previous analyses from the RRID cohort reported independent associations between higher SAF and multiple cardiovascular risk factors including older age, male sex, diabetes, past history of CVD, smoking status, lower eGFR, higher urine protein to creatinine ratio, lower haemoglobin, and lower socioeconomic status [8]. An analysis of deaths after a mean of 3.6 years of observation found that higher SAF was a predictor of ACM in univariable as well as age and sex adjusted models but not in a fully adjusted model [9]. With the benefit of a longer observation period resulting in a greater number of outcome events, we have confirmed that higher baseline SAF and increase in SAF over 1 year are independent predictors of CVEs and ACM in early stage CKD after adjustment for traditional risk factors and, importantly, also CRP. One study has reported that higher SAF predicted incident diabetes, CVEs, and ACM in a large cohort enrolled from the general population [20].

Several mechanisms may account for the association between higher SAF and CVEs as well as mortality. AGEs form cross-links between collagen and elastin molecules in arterial walls resulting in arterial stiffness that has been strongly implicated in the pathogenesis of CVD related to CKD [1]. Additionally, AGEs bind to a specific receptor (receptor for AGE [RAGE]) and provoke endothelial dysfunction [21] as well as inflammation [22] that likely contribute to the pathogenesis of atherosclerosis. Furthermore, in murine models,

atherosclerosis was significantly ameliorated by blockade of RAGE or administration of soluble RAGE [23], suggesting reduced activation of RAGE may prevent atherosclerosis and reduce cardiovascular risk.

Few studies have described longitudinal changes in SAF over time. We found no significant changes in mean SAF over 5 years, but as higher SAF associates with mortality, those with higher baseline levels or a greater increase over time would have been less likely to survive to year 5 follow-up. In multilevel mixed-effects models, we identified multiple baseline variables that were independently associated with higher SAF in repeated measures over the follow-up period including greater age, male sex, diabetes, current or past history of smoking, previous CVD, lower eGFR, lower serum albumin, and lower haemoglobin. These findings are consistent with previous cross-sectional studies that have reported associations between higher SAF and lower GFR as well as other cardiovascular risk factors [8]. Other factors, such as dietary intake and cooking methods, have been associated with changes in AGE levels but were not captured in this population.

SAF is of particular interest as a risk marker, because it is potentially modifiable. A cross-sectional analysis showed lower SAF levels in renal transplant recipients compared to those on either PD or HD, implying that SAF decreases with improved GFR after transplantation [24]. This was confirmed by observation of a decrease in SAF levels in a small number of renal transplant recipients, compared with SAF values recorded while they were on dialysis [24]. AGEs may also enter the body from exogenous sources, including smoking and diet, particularly foods cooked at high temperatures. Dietary changes may therefore also reduce tissue AGE accumulation and SAF measurements. This notion is supported by a cross-sectional analysis of the impact of diet on SAF in persons on HD which reported lower SAF in 27 of 332 participants who followed a vegetarian diet, predicted to be low in AGE content [25]. Furthermore, small randomised studies in persons on HD ($n$ = 18) [26] and PD ($n$ = 20) [27] have reported a reduction in serum AGE levels in response to a low AGE diet, though SAF was not assessed. The hypothesis that dietary AGE restriction is effective to reduce CVEs and improve survival in persons with CKD requires testing in prospective randomised controlled trials.

Sensitivity analyses showed that the association between SAF and CVEs remained present when persons with diabetes were excluded, confirming that elevated SAF was not simply a surrogate for diabetes. Additionally in the multivariable analyses, SAF was an independent predictor of CVEs and ACM but diabetes was not (Tables 3 and 4).

Limitations of this study include a predominantly white and elderly study population. Additionally, the AGE reader is limited in its applicability to persons of African and African-Caribbean ethnicity because of reduced levels of reflected light from darker skin. Our findings may therefore not be applicable to more ethnically diverse or younger populations. Additionally, 1,822 out of 8,280 persons who were invited agreed to participate in the study, potentially resulting in some selection bias. Nevertheless, the baseline data indicate that our study population was representative of patients with CKD followed up in primary care in England [28]. We relied on ICD-10 codes to identify hospital admissions with CVE, and there may therefore have been some misclassification. Nevertheless, coding practice is well-established and rigorous in the NHS, and similar coding data have been used in other large cohort studies including the UK Biobank [29]. At very least, each code represents a hospital admission. Office blood pressure was recorded, but ambulatory blood pressure was not assessed. We were therefore unable to assess the impact of masked hypertension or nocturnal dipping on outcomes. The observed association of lower cholesterol with CVEs and ACM may have resulted from reverse causality due to more persons with CVD receiving lipid lowering therapy. Unfortunately, data on lipid lowering therapy were not available for inclusion in the analysis. Because this was an observational study, the observed associations should not be interpreted as indicating a causal link

between SAF and CVEs or ACM. Prospective trials of interventions that reduce SAF will be required to explore this. Finally, the subgroup analyses should be interpreted with consideration of the fact that in each case a lower number of events resulted in reduced statistical power.

## Conclusions

This analysis showed that higher SAF was independently associated with an increased risk of CVEs and ACM in the largest cohort of persons with CKD stage 3 studied to date. An additional novel finding was that change in SAF over 1 year was associated with an increased risk of CVEs and ACM. These findings support the hypothesis that interventions aimed at reducing AGE levels would be potentially beneficial in improving cardiovascular outcomes and survival in persons with CKD, but this should now be tested in prospective randomised trials.

## Supporting information

**S1 Table. Cox proportional hazards model showing variables associated with time to non-fatal CVEs.** CVE, cardiovascular event.
(DOCX)

**S2 Table. Cox proportional hazards model showing variables associated with time to fatal CVEs.** CVE, cardiovascular event.
(DOCX)

**S3 Table. Cox proportional hazards model showing independent associations with time to first CVE in the subgroup participants who had follow-up assessment of SAF at Year 1 and no CVE prior to Year 1.** CVE, cardiovascular event; SAF, skin autofluorescence.
(DOCX)

**S4 Table. Cox proportional hazards model showing variables associated with time to death due to noncardiovascular causes.**
(DOCX)

**S5 Table. Cox proportional hazards model showing independent associations with time to death from any cause in the subgroup participants who had follow-up assessment of SAF at Year 1.** SAF, skin autofluorescence.
(DOCX)

**S6 Table. Cox proportional hazards model showing independent determinants of time to first CVE in the subgroup participants without Diabetes Mellitus at baseline.** CVE, cardiovascular event.
(DOCX)

**S7 Table. Cox proportional hazards model showing independent determinants of time to death from any cause in the subgroup participants without Diabetes Mellitus at baseline.**
(DOCX)

**S1 STROBE Checklist.**
(DOCX)

**S1 Protocol. Study Protocol Version 2.3, October 2013.**
(DOCX)

**S1 Questionnaire. Study questionnaire.**
(DOCX)

## Acknowledgments

The authors gratefully acknowledge the support of participating GP practices and the participants as well as the essential administrative contributions of Rebecca Packington and Rani Uppal. The authors acknowledge the copyright of the mortality and hospital admissions data provided by the NHS Digital.

## Author Contributions

**Conceptualization:** Adam Shardlow, Natasha J. McIntyre, Richard J. Fluck, Christopher W. McIntyre, Maarten W. Taal.

**Data curation:** Adam Shardlow, Natasha J. McIntyre, Nitin V. Kolhe, Maarten W. Taal.

**Formal analysis:** Natasha J. McIntyre, Nitin V. Kolhe, Laura B. Nellums, Maarten W. Taal.

**Funding acquisition:** Maarten W. Taal.

**Investigation:** Adam Shardlow, Natasha J. McIntyre, Christopher W. McIntyre.

**Methodology:** Adam Shardlow, Natasha J. McIntyre, Nitin V. Kolhe, Richard J. Fluck, Christopher W. McIntyre, Maarten W. Taal.

**Project administration:** Maarten W. Taal.

**Resources:** Maarten W. Taal.

**Software:** Maarten W. Taal.

**Supervision:** Maarten W. Taal.

**Validation:** Adam Shardlow, Natasha J. McIntyre, Nitin V. Kolhe, Richard J. Fluck, Christopher W. McIntyre, Maarten W. Taal.

**Visualization:** Maarten W. Taal.

**Writing – original draft:** Adam Shardlow, Natasha J. McIntyre, Nitin V. Kolhe, Richard J. Fluck, Christopher W. McIntyre, Maarten W. Taal.

**Writing – review & editing:** Adam Shardlow, Natasha J. McIntyre, Nitin V. Kolhe, Laura B. Nellums, Richard J. Fluck, Christopher W. McIntyre, Maarten W. Taal.

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
