## [Editor Report · Decision Letter 0]

12 Feb 2020

Dear Dr Taal, 

Thank you for submitting your manuscript entitled "Skin autofluorescence as a risk factor for cardiovascular events and all-cause mortality in persons with chronic kidney disease stage 3" for consideration by PLOS Medicine.

Your manuscript has now been evaluated by the PLOS Medicine editorial staff and I am writing to let you know that we would like to send your submission out for external peer review.

Kind regards,

Helen Howard, for Clare Stone PhD 

Acting Editor-in-Chief

PLOS Medicine 

plosmedicine.org

---

## [Decision Letter · Decision Letter 1]

20 Mar 2020

Dear Dr. Taal,

Thank you very much for submitting your manuscript "Skin autofluorescence as a risk factor for cardiovascular events and all-cause mortality in persons with chronic kidney disease stage 3" (PMEDICINE-D-20-00405R1) for consideration at PLOS Medicine. 

[LINK]

In light of these reviews, I am afraid that we will not be able to accept the manuscript for publication in the journal in its current form, but we would like to consider a revised version that addresses the reviewers' and editors' comments. Obviously we cannot make any decision about publication until we have seen the revised manuscript and your response, and we plan to seek re-review by one or more of the reviewers. 

We expect to receive your revised manuscript by Apr 10 2020 11:59PM. Please email us (plosmedicine@plos.org) if you have any questions or concerns.

We look forward to receiving your revised manuscript. 

Sincerely,

Adya Misra, PhD

Senior Editor 

PLOS Medicine

plosmedicine.org

Title- Please revise your title according to PLOS Medicine's style. Your title must be nondeclarative and not a question. It should begin with main concept if possible. "Effect of" should be used only if causality can be inferred, i.e., for an RCT. Please place the study design ("A randomized controlled trial," "A retrospective study," "A modelling study," etc.) in the subtitle (ie, after a colon).

Abstract- please provide details of where this study was carried out, dates of patient recruitment/ data collection as appropriate and brief patient demographics

Abstract methods and findings- the last sentence should be a limitation of your study design

The Data Availability Statement (DAS) requires revision. For each data source used in your study: 

STROBE checklist-please do not use page numbers as these are likely to change. Please use paragraphs and sections instead

Please add the following statement, or similar, to the Methods: "This study is reported as per the Strengthening the Reporting of Observational Studies in Epidemiology (STROBE) guideline (S1 Checklist)."

Questionnaires- please provide a copy of all questionnaires used in the study, if previously published please provide a citation. The same goes for the equations mentioned on Pages 6 and 7

Did your study have a prospective protocol or analysis plan? Please state this (either way) early in the Methods section. a) If a prospective analysis plan (from your funding proposal, IRB or other ethics committee submission, study protocol, or other planning document written before analyzing the data) was used in designing the study, please include the relevant prospectively written document with your revised manuscript as a Supporting Information file to be published alongside your study, and cite it in the Methods section. A legend for this file should be included at the end of your manuscript. b) If no such document exists, please make sure that the Methods section transparently describes when analyses were planned, and when/why any data-driven changes to analyses took place. c) In either case, changes in the analysis-- including those made in response to peer review comments-- should be identified as such in the Methods section of the paper, with rationale.

Please replace Caucasian with white 

Please include all financial information into the financial disclosure section within article meta-data instead of the acknowledgements

Comments from the reviewers:

Reviewer #1: I confine my remarks to statistical aspects of this paper. The general approach is mostly fine, but I do have some issues to resolve before I can recommend publication. 

Line 36 (and 162) "Determinants" is too causal. I'm not sure what word to suggest here. Perhaps the editors have a suggestion. But, this is an observational study , so, we only know that these were associated with CVE and ACM.

Line 38 What is after the +- sign? Is that an SD? A 95% CI? or something else?

Line 141 - not a stats comment, but maybe say whether it was the posterior or anterior portion of the forearm? (I only mention this because you discuss skin color and the anterior portion is differently colored than the posterior)

Line 158 - Please specify that it is tertiles of SAF. But why compare across tertiles using ANOVA? it would be better to leave SAF continuous and use regression. (Tertiles might be useful for a table, but categorizing variables for analysis loses power. In addition, nonlinearities can be investigated with splines.

Line 169-170 Linear regression on change scores is not generally recommended, unless the scores are measured perfectly - I'm not sure how well SAF measurements work - a better method is a multilevel model.

Line 170-172 Why were variables log transformed? Linear regression makes no assumptions about the distribution of the variables (it makes assumptions about the residuals).

Figures 2 and 3 - I would use "Years" rather than "Days" just for ease of reading. (People don't think "400 days" they think "a year and a quarter"). I would consider limiting the y axis to 0.5 and up - although this is debatable. It allows finer discrimination but might give the wrong idea to someone who doesn't look at the y axis.

Peter Flom

Reviewer #2: The authors analyze the associations of skin autofluorescence (SAF), a biomarker of tissue accumulation of advanced glycation endproducts, with incidence of cardiovascular events and mortality, in a cohort of 1707 patients affected by stage 3 chronic kidney disease. SAF is a non invasive and quick measurement, which appears to be predictive of adverse health outcomes in particular in diabetic patients. Several studies have evidenced an association of SAF with increased risk for cardiovascular disease and mortality in patients in hemodialysis. However, there are few available data on this association at earlier stages of CKD, who exhibit higher SAF values than the general population. This study thus addresses an important issue, which could help identify specific mechanisms of cardiovascular disease in CKD patients. The methods are sound and the paper is generally well written.

In addition to the analysis of incidence of cardiovascular events and all-cause mortality, the authors should consider adding supplementary analyses regarding cardiovascular and non-cardiovascular mortality. Indeed, it would be informative to differentiate the associations of SAF with fatal versus non fatal cardiovascular events, as well as with mortality from other causes, in order to document the specificity of these associations.

Specific comments:

Title: As "skin autofluorescence" is a rather unspecific term, consider adding "advanced glycation endproducts" in the title.

Abstract: add the number of participants in the methods. 

Statistical methods (page 8, lines 162-168): in the Cox models, SAF was modeled as a continuous variable, thus hypothesizing a linear relationship of SAF with the outcomes. Has this hypothesis been verified ?

Table 1: there is one decimal missing for phosphate in the lowest tertile. A second decimal could also be added for this variable.

Tables 3 and 4: were there any missing data for the covariables ? Please add number of analyzed participants in all analyses.

Discussion (page 21): the authors should also cite the potential selection bias as a limitation of this study, as only 1741 of 8280 invited patients were initially included. This may limit the generalization of the results.

Reviewer #3: This is an important prospective study in a large cohort of patients at an early stage of Chronic Kidney Disease regarding skin autofluorescence as a risk factor for cardiovascular events and all-cause mortality. 

Increased cardiovascular morbidity and mortality in CKD patients is caused by traditional and non-traditional risk factors. The search for new non-traditional CV risk factors, particularly modifiable, which may affect the prognosis in this group of patients, is an important clinical aspect of the study.

My suggestions:

Methods:

1. What kits were used to analyze urine albumin to creatinine ratio?

2. As the medication history was collected at each visit please include the information on how many patients were on lipid lowering therapy?

3. How many patients had malnutrition?

4. Both total and HDL cholesterol were used in analyses. Why was the LDL cholesterol, which increased concentration is a known CV mortality risk factor, and is used as therapeutic goal, not a part of the analyses?

5. Line 169 - BMI abbreviation has not been previously expanded

Results:

1. Line 184 - SBP abbreviation has not been previously expanded

2. Line 185 - DBP abbreviation has not been previously expanded

Discussion:

1. The Authors properly identify that "the association with lower cholesterol may have resulted from reverse causality due to more persons with CVD receiving lipid lowering therapy." However, the most commonly used lipid lowering drugs, statins, except for lowering cholesterol concentrations, possess pleiotropic effects, including decreasing RAGE expression. Therefore statin taking, as a potential confounder, should be a part of the analyses.

2. Please include in limitations that office BP and not ABPM was measured. In CKD patients specific form of hypertension - masked hypertension and non-dipping BP profile, which are known to increase CV risk, are present. The diagnosis is possible only with the use of ABPM.

Acknowledgements:

1. According to PLOS submission guidelines please "Do not include funding sources in the Acknowledgments or anywhere else in the manuscript file. Funding information should only be entered in the financial disclosure section of the submission system."

Reviewer #4: This is a well designed prospective study performed by an experienced team in the field. In fact, it is a continuation of an already published work by these authors. It adds on additional information of the role of AGEs on the clinical outcome of the patients with Chronic kidney disease stage 3. There is also some new information in this paper: the finding that over time the accumulation of AGEs in the body rises, with an implication that the risk for development of cardiovascular events or all cause mortality rises too. It also points out the direction of future research in this field, designing prospective interventional clinical studies with an intention to reduce AGEs and consequently the adverse clinical outcomes in the patients. The paper is clearly and critically written. It is a significant contribution to the existing literature. I would like to congratulate the authors for the excellent work and gladly recommend publication.

[LINK]

---

## [Decision Letter · Decision Letter 2]

13 May 2020

Dear Dr. Taal,

Thank you very much for re-submitting your manuscript "The association of skin autofluorescence, a measure of advanced glycation endproduct accumulation, with cardiovascular events and all-cause mortality in persons with chronic kidney disease stage 3: a prospective cohort study" (PMEDICINE-D-20-00405R2) for review by PLOS Medicine.

I have discussed the paper with my colleagues and the academic editor and it was also seen again by xxx reviewers. I am pleased to say that provided the remaining editorial and production issues are dealt with we are planning to accept the paper for publication in the journal.

[LINK]

We look forward to receiving the revised manuscript by May 20 2020 11:59PM. 

Sincerely,

Adya Misra, PhD

Senior Editor 

PLOS Medicine

plosmedicine.org

Requests from Editors:

Title- Consider shortening to “The association of skin autofluorescence with cardiovascular events and all-cause mortality in persons with chronic kidney disease stage 3: a prospective cohort study”

Abstract

Please provide brief participant demographics

Last sentence of methods and findings must state a limitation of your study design/methodology 

Competing Interests- please include a note to state that Maarten Taal is an Academic Editor at PLOS medicine

Data availability- please can you update the meta-data section with the statement provided in the response to comments? It seems the original data statement was not updated, we would be grateful if you could change this. 

Author Summary

Would it be possible to combine the last 3 points in the “why was this done” section to two bullet points? Perhaps combine the third and fourth points?

In the “what did the researchers do and find” section, we only need the key messages. I suggest combining the second and third point. The last point can perhaps be removed since you mention this again in the “what do these findings mean” section. 

Results

Line 235- please replace “diabetic” with “with Type 1/2 diabetes” 

Please format the bibliography using Vancouver style

Individual items in the supplementary information like the STROBE checklist, study protocol and questionnaires etc will need to be dissociated and uploaded as individual files.

Comments from Reviewers:

Reviewer #1: The authors have addressed my concerns and I now recommend publication

Peter Flom

Reviewer #2: The authors have adequately answered most of the reviewers' comments. I have only two additional comments:

- I don't think that Figures 2 and 3 allow to correctly assess the linearity of the association of SAF with ACM and CVE. I think that the authors should test this hypothesis, for instance by using splines as previously suggested by reviewer #1.

- Regarding cholesterol, I understand that LDL cholesterol can only be deducted from total and HDL-cholesterol. However, as total includes HDL-cholesterol, I think that it would be more correct to analyse HDL- and non HDL-cholesterol, rather than HDL and total cholesterol (also because non HDL cholesterol is one of the strongest biomarker of cardiovascular disease in the general population as shown for instance by Brunner FJ et al. Lancet 2019 394(10215):2173-2183). I admit that this is a detail, since cholesterol does not appear to be associated with CVE or ACM in these CKD patients. 

Reviewer #3: I would like to congratulate the authors for the excellent work.

Reviewer #4: I examined the revised version. Earlier, I recommended accept. This version looks fine.

[LINK]

---

## [Editor Report · Decision Letter 3]

11 Jun 2020

Dear Prof Taal, 

On behalf of my colleagues and the academic editor, Dr. Cecile Delcourt, I am delighted to inform you that your manuscript entitled "The association of skin autofluorescence with cardiovascular events and all-cause mortality in persons with chronic kidney disease stage 3 : a prospective cohort study" (PMEDICINE-D-20-00405R3) has been accepted for publication in PLOS Medicine. 

PRODUCTION PROCESS

PRESS

PROFILE INFORMATION

Thank you again for submitting the manuscript to PLOS Medicine. We look forward to publishing it. 

Best wishes, 

Adya Misra, PhD

Senior Editor 

PLOS Medicine

plosmedicine.org